# Updated Information of the Effects of (Poly)phenols against Type-2 Diabetes Mellitus in Humans: Reinforcing the Recommendations for Future Research

**DOI:** 10.3390/nu14173563

**Published:** 2022-08-30

**Authors:** Regina Menezes, Paulo Matafome, Marisa Freitas, María-Teresa García-Conesa

**Affiliations:** 1CBIOS—Universidade Lusófonás, Research Center for Biosciences & Health Technologies, Campo Grande 376, 1749-024 Lisboa, Portugal; 2NOVA Medical School—Faculdade de Ciências Médicas (NMS—FCM), Universidade Nova de Lisboa, Campo dos Mártires da Pátria 130, 1169-056 Lisboa, Portugal; 3iBET, Instituto de Biologia Experimental e Tecnológica, Apartado 12, 2781-901 Oeiras, Portugal; 4Coimbra Institute of Clinical and Biomedical Research (iCBR) and Institute of Physiology, Faculty of Medicine, University of Coimbra, 3000-548 Coimbra, Portugal; 5Center for Innovative Biomedicine and Biotechnology (CIBB), University of Coimbra, 3000-548 Coimbra, Portugal; 6Clinical Academic Center of Coimbra, 3000-548 Coimbra, Portugal; 7Instituto Politécnico de Coimbra, Coimbra Health School (ESTeSC), Department of Complementary Sciences, 3000-548 Coimbra, Portugal; 8LAQV-REQUIMTE, Laboratory of Applied Chemistry, Department of Chemical Sciences, Faculty of Pharmacy, University of Porto, 4050-313 Porto, Portugal; 9Research Group on Quality, Safety and Bioactivity of Plant Foods, Centro de Edafología y Biología Aplicada del Segura (CEBAS), Spanish National Research Council (CSIC), Campus de Espinardo, 30100 Murcia, Spain

**Keywords:** blood glucose, diabetes, glycated hemoglobin, interindividual variability, polyphenols, metabolites, bioavailability

## Abstract

(Poly)phenols have anti-diabetic properties that are mediated through the regulation of the main biomarkers associated with type 2 diabetes mellitus (T2DM) (fasting plasma glucose (FPG), glycated hemoglobin (HbA1c), insulin resistance (IR)), as well as the modulation of other metabolic, inflammatory and oxidative stress pathways. A wide range of human and pre-clinical studies supports these effects for different plant products containing mixed (poly)phenols (e.g., berries, cocoa, tea) and for some single compounds (e.g., resveratrol). We went through some of the latest human intervention trials and pre-clinical studies looking at (poly)phenols against T2DM to update the current evidence and to examine the progress in this field to achieve consistent proof of the anti-diabetic benefits of these compounds. Overall, the reported effects remain small and highly variable, and the accumulated data are still limited and contradictory, as shown by recent meta-analyses. We found newly published studies with better experimental strategies, but there were also examples of studies that still need to be improved. Herein, we highlight some of the main aspects that still need to be considered in future studies and reinforce the messages that need to be taken on board to achieve consistent evidence of the anti-diabetic effects of (poly)phenols.

## 1. Type 2 Diabetes Mellitus: Current Scenario

Type 2 diabetes mellitus (T2DM) is globally on the rise. The International Diabetes Federation estimates that >700 million people will be affected with T2DM by 2045 [1]. T2DM is a complex metabolic pathology commonly related to body weight adiposity concomitant with age, reduced physical activity, and unhealthy diets. Obesity is typically associated with ectopic fat deposition and lipotoxicity in insulin-sensitive peripheral tissues (liver, skeletal muscle, adipose tissue), which can result in the activation of oxidative stress and inflammatory pathways that ultimately inhibit the insulin receptor [2]. As a consequence, these tissues slowly develop resistance to the normal response to insulin (insulin resistance, IR), and pancreatic β-cells are stimulated to produce higher amounts of insulin to compensate for this systemic resistance and maintain normal glycaemia (<100 mg/dL). This situation can last for decades, but β-cell exhaustion progressively leads to lower insulin secretion and an increase in fasting plasma glucose (FPG), going from an intermediate state referred to as prediabetes (100–125 mg/dL) to T2DM (≥126 mg/dL) [3] (Figure 1). The diagnosis and treatment of T2DM is primarily based on the circulatory levels of FPG, glycated hemoglobin (HbA1c), 2 h glucose after an oral glucose (75 g) challenge (OGTT), and a mean daily glucose measured through continuous monitoring. HbA1c reflects the average level of blood glucose over 3 months, which is the gold standard indicator for monitoring blood glucose and is currently considered the main biomarker for the long-term control of T2DM [4]. The criteria for the screening and diagnosis of diabetes, as well as of prediabetes as updated by the American Diabetes Association (ADA) [3], are summarized in Figure 1.

Long-term untreated hyperglycemia can lead to serious pathologies, including vascular and neuronal complications and cardiovascular diseases (CVDs) [5]. Currently, the most effective strategy to promote the remission of hyperglycemia and prevent T2DM is to induce substantial weight loss by implementing lifestyle changes (calorie restriction, healthy foods, increased physical activity), and in cases of severe obesity, through metabolic surgery [6,7]. Alternatively, different anti-diabetic drugs approved by the Food and Drug Administration (FDA) [8] (e.g., biguanides (metformin), inhibitors of glucose absorption (SGLT2 inhibitors, acarbose), glitazones (PPARγ agonists), modulators of incretin (DPP-IV inhibitors and GLP-1 receptor agonists), insulin secretagogues (sulphonylureas), etc.) can be used alone or in combination to moderate the progression of T2DM. These drugs act by controlling the glycemic index, reducing HbA1c and slowing the progression of vascular complications. Nevertheless, these therapeutic goals are not always achieved, and the outcomes can be highly variable between individuals. For example, the pharmacological reduction in HbA1c may vary from ≈0.3% up to nearly 2% depending on the baseline values and the type of drug [8]. In this scenario, the search for new and more effective regulators of IR and T2DM development remains a continuous aim of research.

## 2. (Poly)phenols: A Brief Overview

Plants constitute a major source of natural bioactive molecules. Among those, (poly)phenols are phenolic compounds with single or multiple hydroxyl (OH) groups that are devoid of any nitrogen-based functional group and range from simple phenolics to complex high-molecular-weight polymers. (Poly)phenols constitute a group of at least 10,000 known different molecules that can be classified into different families (Figure 2): phenolic acids, with compounds derived from hydroxybenzoic acids (e.g., gallic acid), and from hydroxycinnamic acid (e.g., caffeic acid, ferulic acid, coumaric acid); stilbenes (e.g., resveratrol); flavonoids, including flavonols, flavones, isoflavones, flavanones, anthocyanidins, and flavanols; and lignans (e.g., secoisolariciresinol) [9].

(Poly)phenols can be found in plant foods and beverages such as fruits, vegetables, cereals, coffee, and tea [10]. It has been estimated that the intake of these compounds in Europe, mainly flavonoids and phenolic acids from fruits, chocolate and vegetable juices, can vary from 167 to 564 mg/day [11]. This intake might considerably increase, reaching g quantities, by the consumption of nutraceuticals and enriched products. (Poly)phenols have long been associated with many health benefits. Due to their reactive chemical characteristics, these compounds have the capacity to interfere with the metabolism and molecular responses of cells and have been investigated for their protective role against cancer, inflammatory and neurodegenerative diseases, metabolic disorders, and CVDs [12]. Importantly, the (poly)phenol structure and its metabolic transformation during the digestion and absorption determine their bioavailability and in vivo interactions with cells and biomolecules, consequently determining their bio-efficacy [13]. Somewhere across the nutritional and pharmacological strategies, (poly)phenols have long been investigated for their ability to modulate T2DM-associated biomarkers and influence a range of cellular and molecular mechanisms implicated in the development of IR and hyperglycemia [5]. However, despite the large number of published pre-clinical and clinical studies supporting these regulatory effects, there are not yet real and fully substantiated recommendations for any specific (poly)phenol or (poly)phenol-containing products that could be effectively applied as a means to ameliorate high levels of Hb1Ac, FPG, and associated metabolic alterations.

The main objectives of this article are: (i) to update and summarize the current knowledge of the benefits of dietary (poly)phenols against T2DM in humans; (ii) to re-examine and evaluate the main limitations and problems of the accumulated pre-clinical evidence (in vitro and animal testing) supporting the role and potential mechanisms of action of (poly)phenols against T2DM; (iii) to highlight the main limitations and problems of current clinical intervention trials and pre-clinical studies and reinforce some of the strategies that will help to direct future research, improving the evidence for the anti-diabetic benefits of these compounds and supporting their nutritional and (or) medical recommendations against T2DM.

## 3. Updated Status of the Evidence of the Benefits of (Poly)phenols against T2DM in Humans

### 3.1. Results from Recent Systematic Reviews and Meta-Analyses: What Are the Main Messages?

A very recent prospective analysis in the context of the PREDIMED-Plus trial investigated the association between the intake of (poly)phenols and the levels of HbA1c and blood glucose in a sample population of overweight/obese participants with metabolic syndrome (MetS). The authors concluded that an increment in the intake of some classes of (poly)phenols was inversely associated with reduced levels of these two key biomarkers. The authors also reported a different response to the (poly)phenols depending on the diabetes status and indicated that the greatest benefit was observed in individuals with pre-diabetes [14]. However, a closer look at the data shows highly variable results depending on the different statistical models applied, and thus it is possible to find significant associations between the reduction in these biomarkers and the increase or decrease in the intake of some (poly)phenols. Within the subgroup of individuals with pre-diabetes, some (poly)phenols modified glucose levels but not HbA1c and vice versa, and the changes after one year were generally rather small. Even though this study collects data from a rather large sample population, the complexity of the dietary changes promoted during the PREDIMED intervention trial and of the many confounding factors that may influence the results makes it difficult to interpret the variable outcomes, which do not clearly support the benefits of these compounds against T2DM. Indeed, the authors recognize the need to investigate all these factors and to truly demonstrate the effects of these compounds through properly designed randomized clinical trials (RCTs).

We went through some of the most recent reviews and meta-analyses gathering different RCTs testing the antidiabetic effects of different (poly)phenols [5,15,16,17,18]. The main evidence refers most commonly to changes in FPG and HbA1c, while the effects on other related variables such as HOMA-IR or insulin are less frequently indicated. Additionally, the majority of the RCTs were conducted with mixtures of (poly)phenols, i.e., flavanols from cocoa, anthocyanidins from berries, and tea catechins. As for single compounds, resveratrol is one of the most investigated natural plant phenolics for their potential regulatory effects against T2DM.

Raimundo et al. [15] gathered a total of 20 RCTs carried out with different types of (poly)phenols and reported a general antidiabetic effect of these compounds by significantly reducing both FPG and HbA1c. Nevertheless, the overall effects were small (MD: −3.32 mg/dL for FPG and −0.24% for the glycated hemoglobin), and the heterogeneity of the studies was rather high (I^2^ > 50%). When looking at subgroups of specific compounds, such as flavanols or resveratrol, the effects on glucose levels were limited to a lower number of studies and became insignificant. One additional outcome was that the effects on glucose appeared to be more substantial in individuals with diabetes taking anti-diabetic medication. The main limitations indicated by the authors were an insufficient number of RCTs and the lack of sufficient data and information to allow for the analysis of specific subgroups. With regard to analyses in more specific subpopulations, the systematic review by Sánchez-Martínez L. et al. [16] analyzed the effects of mixed (poly)phenols in a population of post-menopausal women. The results indicate a lack of significant effects of these compounds on the levels of glucose with the effect size varying from −17.0 mg/dL to +12.0 mg/dL. Of the 13 trials included, only two of them reported a significant reduction in FPG and insulin following the consumption of a green tea extract, rich in flavan-3-ols. The authors widely discussed several issues related to the quality of studies and the factors that affect interindividual variability, which must be addressed in future studies to validate the effects of (poly)phenols against T2DM in specific and well-characterized individuals. The effects of (poly)phenols in on glycemic biomarkers in a population of individuals with diabetes characterized by a developed nephropathy have also recently been reviewed [17]. The authors collected 13 trials and reported a general non-significant effect in FPG (+2.78 mg/dL) but a significant mean reduction in HbA1c (−0.27%). It is interesting to note that of all the RCTs investigated in this study, only the RCT carried out with resveratrol reported a significant effect on HbA1c, while the rest of the studies conducted with other (poly)phenols reported no effects on this biomarker. Since, in general, the studies were categorized as low or very low quality and with a high risk of bias, the results of this latest study highlight the problems of conducting meta-analyses with this type of low-quality studies, which may lead to misleading conclusions. In a more recent review, Fernandes et al. [5] re-examined a compendium of pre-clinical and clinical studies to support the anti-diabetic properties of nutraceuticals containing (poly)phenols. They summarize a few examples of human trials looking at the reducing effects of resveratrol, curcumin, flavanols, and anthocyanidins on FPG. However, the strength of the evidence of these results was not presented or discussed, and the final message of the review was that more and larger RCTs were needed, and that the interindividual variability and factors involved need to be investigated. All these previous reviews and meta-analyses collected and summarized the antidiabetic effects of mixed (poly)phenols from different food sources. We also revised the particular case of (poly)phenols present in tea, a beverage widely investigated for its multiple metabolic benefits. In the latest meta-analysis [18], covering a total of 27 RCTs, the pooled results show that green tea significantly lowered FPG (−1.44 mg/dL; *p* < 0.001) with very low heterogeneity (I^2^ = 7.7%) but did not significantly modify HbA1c (−0.06%) and insulin values. The conclusion of this meta-analysis was that long-term trials assessing the effects of green tea supplementation on glycemic control are still needed. Several reviews supporting the antidiabetic effects of the bioactive constituents of tea and their potential mechanisms of action (i.e., modulated signaling pathways, microbiota interaction) have been recently published [19,20,21]. These studies conclude that the current evidence of the effects of tea and tea (poly)phenols against T2DM remains limited and inconsistent and corroborate the need for further intervention trials.

We also revised the results of recent metanalyses collecting evidence of resveratrol, as an example of a single (poly)phenol compound broadly investigated to combat T2DM. The results and messages of the revised articles are varied. The study carried out by Jeyaraman et al. [22] led to a total of only three RCTs found in individuals with diabetes, with no significant and very small effects on HbA1c (0.1%), FPG (2.0 mg/dL) and IR (−0.35). Although this revision was conducted following all specifications required by the Cochrane database of systematic reviews, the limited number of studies included prevented any conclusion regarding the effects of resveratrol. In the same year, Nyambuya et al. [23] published another meta-analysis on the impact of supplementation with resveratrol in T2DM patients following hypoglycemic therapy. The authors collected up to five RCTs and reported a message of lower levels of FPG and insulin with resveratrol. However, the data did not fully support this finding since the results were again rather small and not significant (FPG SMD: −0.06; insulin SMD: −0.08; Hb1Ac SMD: +0.18). In both cases, the heterogeneity of the studies was rather high (I^2^ up to 73%). In a more recent meta-analysis, Delpino and Figueiredo [24] collected and analyzed a total of 30 studies. In this case, the participants were patients with different metabolic disorders and the overall reported results were that resveratrol had significant effects on IR (SMD: −0.34) and Hb1Ac (SMD: −0.64) and that blood glucose was significantly reduced only in individuals with diabetes (SMD: −0.85). Once more, the heterogeneity was rather large (I^2^ ≈ 70% to 90%). Furthermore, Gu et al. [25] also published a systematic review and meta-analysis of the effects of resveratrol on various metabolic biomarkers in patients with T2DM, including a total of 19 RTCs. Following the analyses, the authors reported an overall non-significant effect of resveratrol on glucose levels, but the effects became significant at the higher doses of the compound (>1.0 g/d, MD: −18.76 mg/dL).

These results show the disparity in the outcomes and messages conveyed by the most recent systematic reviews and meta-analyses regarding the beneficial effects of (poly)phenols against T2DM. Although some of the accumulated evidence suggests some potential regulatory benefits, the data remain limited and contradictory. The overall effects appear to be rather small and highly variable and are very influenced by the small number, high heterogeneity and low quality of the RCTs examined in those meta-analyses. As it has been repeatedly stated, one of the main building blocks used to achieve better evidence of the efficacy of (poly)phenols against chronic metabolic disorders is the design and implementation of the best-designed human RCTs. These studies need to incorporate a number of essential characteristics with regard to the number and phenotypic homogeneity of the participants, test products and placebo, and most importantly, the metabolic fate (bioavailability) of the test compound/s so that, ultimately, these studies can provide more definitive evidence for the regulatory effects of (poly)phenols against T2DM [5,15,26,27].

### 3.2. Results from Very Recent RCTs: Are We Improving RCTs Design to Achieve Better Evidence for the Effects (Poly)phenol against T2DM?

A quick literature search in PubMed^®^ (https://pubmed.ncbi.nlm.nih.gov last accessed on the 27 July 2022) using the search terms “polyphenol” and “diabetes” filtered for RCTs and published in 2022 retrieved nine studies that investigated the potential regulatory effect of (poly)phenols and (poly)phenols-containing products on T2DM. We critically examined the reported effects on the main biomarkers of T2DM in chronic interventions and evaluated whether they improved the trial design by implementing some of the features indicated above.

We came across some new studies looking at the potential anti-diabetic effects of extracts from different plants and seaweeds containing mixed (poly)phenols. An interesting and well-designed RCT was conducted to investigate the protective action of *Moringa oleifera* dry leaf powder (2.4 g/day for 12 weeks), with a high (poly)phenol content (2300 mg/Kg) in individuals with prediabetes [28]. The results show significant differences in the levels of FPG and HbA1c between the intervention group (*N* = 31) and the placebo (*N* = 34), which led the authors to suggest an antihyperglycemic activity of the *Moringa oleifera* dry leaf powder. In their study, the authors attempted to increase the sample size per arm and select a specific group of individuals with pre-diabetes of a similar age and body weight who were not taking antidiabetic medication, providing a more homogeneous sample population. However, they still present the overall results for a mixed population of men and women, and the extract contained many other compounds (minerals, carbohydrates, etc.) precluding the attribution of the antidiabetic effects to the (poly)phenols present in the extract. Furthermore, it is important to consider and discuss the size of the reported effects after 12 weeks of the intake of this product in the context of T2DM. A small reduction in the glucose levels (4–5 mg/dL) might be important in individuals with pre-diabetes if this effect can be sustained in the long term. On the other hand, the reported 2% reduction in glycated hemoglobin is a substantial response in comparison with the values attained with drugs (0.3–2.0%). In another study published in the same year, Grabez et al. [29] reported the effects of a pomegranate peel (poly)phenol-containing extract in capsules (0.5 g/day for 8 weeks) in an adult mixed population of T2DM men and women. The authors reported an improvement of HbA1c in the intervention group (*N* = 31) as compared to the control group (*N* = 34) (7.55 ± 1.22 vs. 7.32 ± 1.02%, *p* < 0.001) and suggested a potential effect to mitigate altered biomarkers in the context of T2DM. Nevertheless, this reduction, although statistically very significant, was rather small (≈0.2%), and the product did not have an effect on glucose. Once more, the test product contained a mixture of compounds making the attribution of the observed properties to the (poly)phenols impossible. One additional study evaluated the potential of a brown (poly)phenol-rich seaweed extract (*Ascophyllum nodosum* and *Fucus vesiculosus*; 0.5 g/day for 12 weeks) in combination with a low-calorie diet in overweight/obese individuals with prediabetes [30]. Although some inflammatory markers were attenuated, the authors did not detect significant changes in blood glucose, insulin levels or glycated hemoglobin in the brown seaweed intervention group (*N* = 27) when compared with the placebo group (*N* = 29). As mentioned in the previous section, resveratrol remains in the spotlight as a potential molecule conferring a range of benefits, including some potential effects against T2DM. Nonetheless, these properties have not yet been fully demonstrated. This year, Mahjabeen et al. [31] reported that the intake of pure (99%) resveratrol (0.2 g/day for 24 weeks) significantly decreased the levels of FPG (−9.0 mg/dL), Hb1Ac (−0.45%), insulin (−1.31 mUI/L) and HOMA-IR (−0.83) in T2DM patients (*N* = 55 treatment group, *N* = 55 placebo group) taking oral anti-hyperglycemic agents, providing further support to this molecule as a potential co-adjuvant against T2DM.

The studies included here present additional and interesting results that contribute to the knowledge on the potential impact of (poly)phenols on T2DM. However, despite the accumulated messages on the specific issues that need to be incorporated into this type of intervention, we still found some aspects that need to be improved: (i) the sample size per arm remains limited in many studies; (ii) the sample participants, although focused on patients with prediabetes or T2DM were still a mixed population of men and women, with variable body weights, ages and (or) medications, all of which have a high and variable impact on the results; (iii) the use of g quantities of mixed compounds as the test product makes the attribution of the effects to the (poly)phenols rather difficult; and (iv) the reported effects remain small or not significant in several cases sustaining doubts regarding the efficacy of consuming some of these products as a means of preventing T2DM. Products such as resveratrol and the *Moringa oleifera* extract appear to offer a good anti-diabetic potential but they still need further research. It is important to find out whether antidiabetic benefits can be sustained in the long term (months to years) without any toxic or adverse effects and/or to confirm whether, in the case of the extract, the reduction in blood glucose or HbA1c are truly caused by the intake of any (poly)phenols.

Furthermore, despite the growing knowledge on the metabolic transformation and fate of (poly)phenols, there are only a few studies that have jointly investigated the metabolism and bioavailability of these compounds while evaluating their beneficial effects in humans. This is important since the action of the (poly)phenols may be partially mediated by the interaction between their metabolites and different target cells and biomolecules in the organism. Along these lines, a recent pilot study conducted by Moreno et al. [32] constitutes a good example of an experimental design aiming to show the protective role of some (poly)phenols towards diabetes and identify the metabolites that may be implicated. To achieve this goal, the authors provided an oral dose of red raspberry (*Rubus*
*idaeus*) (123 g/day for 2 weeks) to T2DM patients and monitored blood samples at baseline and post-feeding periods for IR biomarkers, as well as for the presence of phenolic metabolites. The authors reported that the red raspberry intervention was associated with a downward trend in HOMA-IR and identified several (poly)phenol metabolites with the potential to reach internal tissues. One of these, 3,4-dihydroxyphenylacetic acid, was also tested for its capacity to stimulate insulin secretion in an in vitro model of pancreatic β-cells. In a similar fashion, the study conducted by Duarte et al. [33] also shows that the intake of a *Passiflora setacea* juice containing mixed (poly)phenols had reducing effects on insulin and HOMA-IR. The authors extracted some of the plasma-sulfate- and glucuronide-derived metabolites, which were subsequently used to treat microglial cultured cells and showed some regulatory activity in biomarkers potentially related to anti-diabetic effects. In spite of the limitations of the results of these studies, they represent an experimental design aiming to elucidate how the intake of a product rich in (poly)phenols may attenuate the development of diabetes via generated metabolites and constitute a good example to follow to further improve future studies in this area.

## 4. How Can Future Pre-Clinical Studies Be Improved to Further Support the Evidence of (Poly)phenols against T2DM?

A plethora of pre-clinical studies have long supported the beneficial properties of dietary (poly)phenols. However, it has also been long recognized that many of those studies were conducted under experimental conditions with limited transferability to real in vivo conditions, posing reasonable doubts regarding the effectiveness of the compounds to fight against diseases in humans. In the following sections, we revise and discuss some recent cell and animal studies looking at the antidiabetic properties of (poly)phenols to try to highlight some of the main issues that still need to be improved.

### 4.1. In Vitro Testing Using Cell Cultures and Cell-Free Assays

Most in vitro cell culture studies have been carried out with the objective of investigating cell responses and molecular mechanisms triggered in different cell models following exposure to specific (poly)phenols or products containing mixed (poly)phenols. Those studies have led to describing different potential means by which (poly)phenols may interfere with the development of IR and T2DM, including the regulation of glucose transport and absorption, or the modulation of the expression and activity of key metabolic, oxidative stress, and inflammatory biomarkers in cell models of the liver, intestine, adipose tissue, muscle, pancreas, brain, or vasculature [5,34]. However, as already stated, many of those studies do not always represent a real biological situation since the experimental conditions cannot be typically achieved in vivo.

Many of the critical issues that need to be considered to improve the design of cell culture studies have been repeatedly indicated [35,36] but appear to still be overlooked in many published in vitro studies looking at the effects of (poly)phenols. One of the essential issues to consider is the metabolic transformation and bioavailability of the parent ingested (poly)phenols since the compounds that should be tested against model cells are the specific metabolites formed in vivo, that circulate in blood and may be able to reach the specific target tissues. In addition to the hepatic metabolism, exposure of the (poly)phenols to gut microbiota has been shown as the main mechanism of the transformation and degradation of these compounds. Equally important is the issue of the concentration of those metabolites. After oral administration, the serum levels of (poly)phenols are generally very low due to the rapid metabolism, low absorption, and rapid systemic elimination and clearance, which drastically reduce the concentration available for the therapeutic effects [37]. Manach et al. [38] reviewed the bioavailability of the main dietary (poly)phenols and reported that the concentration in the plasma of some the main derived metabolites was in the nM to low µM range. These values have not been frequently used in the experimental models; instead, they are more often carried out with higher concentrations. The quantitative and qualitative determination of the host and microbial metabolites that form in vivo following the intake of (poly)phenols constitute a main area of research that should be further developed to enhance our knowledge and understanding of the impact of dietary (poly)phenols against metabolic diseases including T2DM [39].

We searched for recent cell culture studies on the modulatory antidiabetic effects of (poly)phenols and found some examples of studies looking at the effects of the metabolites that can be found in plasma following the intake of (poly)phenols. Zhang et al. [40] investigated the antidiabetic effects of urolithin A in β pancreatic cells, where this microbial metabolite derived from ellagitannins was able to modulate a number of inflammatory, metabolic and stress biomarkers. The concentration of the metabolite used in the experiments was, however, rather high (50 µM). More recently, García-Diez et al. [41] published their study looking at the effects of the microbial-derived flavanol metabolite 2,3-dihydroxybenzoic acid on several key regulatory mechanisms related to the energy metabolism balance and insulin signaling in cardiac cells. The authors reported the modulation of various markers, including an increase in the expression of the glucose transporter GLUT-4. In this exercise, the concentrations used were more in consonance with potential circulating values of the metabolite (1 to 10 µM). Nonetheless, we were recently able to find examples of a poorer experimental design [42]. In this case, the authors reported the hypoglycemic effect of a leaf extract from *Origanum vulgare* through the inhibition of α-glucosidase activity, promotion of glucose uptake, inhibition of glycosylation and relieving of oxidative stress in liver cells exposed to µg/L of the extract. It is difficult to envisage how hepatic cells will ever be exposed to this or similar extracts under in vivo conditions. Overall, better-designed cell culture studies are still needed to support the potential antidiabetic properties of dietary (poly)phenols and their mechanisms of action. Some of the main problems that still need to be solved are the complete identification and quantification of all the metabolites that can be truly found in the human tissues following the intake of (poly)phenols. Equally important is to develop methods to synthesize and provide pure and sufficient quantities of all these molecules so they can be widely and reliably tested in the cell models. Furthermore, experimental designs should ideally combine and simultaneously test the different metabolites at a nM—low µM scale in the different insulin-sensitive cell models represent the tissues implicated in the development of T2DM [43].

In addition to cultured cells, cell-free assays have also been widely used to investigate the effects of (poly)phenols on enzymes directly involved in T2DM pathophysiology. The inhibition of the activity of key enzymes such as amylases and glucosidases (carbohydrates digestion) or glycogen phosphorylase (a key enzyme in the glycogenolysis pathway) are considered additional potential mechanisms by which dietary (poly)phenols can interfere with the development of diabetes. These studies, although simple and informative, also have intrinsic difficulties that need to be carefully considered to increase the validity and interpretation of the results. The list of issues to take into account is long and includes the selection of the source of enzyme (human vs. non-human); the incubation conditions (time, pH, temperature); the nature of the substrates (natural vs. synthetic); the comparison to appropriate positive controls (e.g., acarbose); the solubility of the test compounds/extracts, which often determines the use of organic solvents that may influence the enzyme activity; the use of colorimetric and fluorometric methods that can be interfered by the absorbance or quenching effects of the (poly)phenols; etc. All these factors can lead to falsely attributed anti-diabetic properties in some (poly)phenols and focus subsequent investigations on the wrong product/compound. It is thus important that these assays are somehow normalized and performed following common recommendations and conditions. Along these lines, the study published by Rocha et al. [44] reports the optimization and validation of an in vitro standardized glycogen phosphorylase activity assay that can be applied to panels of structurally related (poly)phenolic compounds under the same experimental conditions. This protocol allows for the establishment of accurate structure–activity relationships and the identification of (poly)phenols with sufficient evidence to support their antidiabetic potential.

### 4.2. Animal Models

The use of animal models of T2DM remains important for the investigation and understanding of this complex disease, as well as for the discovery of new and more effective means of treatment. Different animal models of diabetes have been developed and employed including chemically induced, genetically induced, diet-induced and obesity-associated models [43,45]. (Poly)phenols and (poly)phenol-containing products have been investigated for their antidiabetic properties in hundreds of these animal studies, principally in models of small rodents (rats and mice), and this number is continuously growing. In a very recent review, Niewiadomska et al. [46] collected the metabolic regulatory properties for different single (poly)phenols (resveratrol, curcumin, quercetin) and (poly)phenol-containing products (red wine, cocoa, green tea, cinnamon, pomegranate) in various rat models of metabolic disorders (diabetic Zucker rats, hypertensive rats, diet- and/or chemically induced Wistar and Sprague Dawley rats). The results show that the intake of these compounds promotes multiple metabolic effects, including the regulation of glucose and insulin, as well as the modulation of a range of biomarkers implicated in metabolism, inflammation and oxidative stress, encouraging the testing of these compounds to combat T2DM in humans.

Animal studies reveal critical information that cannot be attained in human studies. In this regard, animal models allow for the investigation of responses triggered in a wide range of target tissues that may constitute, at least in part, the mechanisms of action that underlay the antidiabetic properties of (poly)phenols. Additionally, animal studies can be effectively used to compare dietary and pharmaceutical treatments of T2DM. Along these lines, Boonphang et al. [47] reported that an aqueous extract of *Coffea arabica* pulp was able to reduce the levels of plasma glucose and the HOMA index in a rat model of T2DM, similar to the effects attained with metformin. These researchers also investigated the regulatory effects of the extract on a number of biomarkers, including the expression of several antioxidant genes and the organic cation transport function in the renal tissue of the animals.

Despite the metabolic differences between species, animal models also provide essential information on the metabolic fate and tissue distribution of (poly)phenols that can contribute to the understanding of human metabolism. Animal studies allow for the qualitative and quantitative analysis of (poly)phenol-derived metabolites that can be detected in different target tissues, providing evidence of the metabolic fate of these compounds. These metabolic studies in connection with the study of the molecular responses in those same tissues can help to picture the mechanisms of action underlying the antidiabetic properties of the (poly)phenols. A recent study conducted in high-fat diet (HFD)-induced obese mice reported that a (poly)phenol-rich extract of *Antirhea borbonica* significantly reduced glycemia, insulinemia, OGTT and the HOMA-IR [48]. The authors identified some of the circulatory-derived metabolic derivatives (i.e., caffeic acid and the microbial metabolite hippuric acid), and further confirmed similar antidiabetic effects for caffeic acid. In addition, the investigators looked at the potential regulatory mechanisms of this compound in murine bEnd3 cerebral endothelial cells, where they detected some downregulatory effects on specific matrix metalloproteinases (MMP-2) associated with hyperglycemia. More animal studies looking at the combined analysis of diabetic biomarkers, metabolic fate, and target tissue responses will enhance our knowledge and understanding of the potential of (poly)phenols as a means to prevent T2DM. One additional and important issue that should be clearly addressed in animal studies is the dose of (poly)phenols and whether the human equivalent dose (HED) is achievable through diet or by the intake of reasonable quantities of a particular (poly)phenol-containing product. In the two examples above, the estimated HED [49] was around 200 and 250 mg/day for a 70 Kg individual, which is a reasonable amount to consume.

Establishing the adequate dose of (poly)phenols needed to prevent diseases is also very important with regard to the potential toxicity of these compounds. It has been estimated that a dose of (poly)phenols provided through the diet can reach several hundred mg and even up to g amounts in human populations with a high intake of (poly)phenol-containing foods and beverages [50]. Nevertheless, the evidence of the potential toxicity of the long-term consumption of high doses of (poly)phenols is not yet known. The findings from cell and animal toxicity studies remain controversial, and the results are highly variable depending on the model and type of cells used as well as the type of compounds [51]. Since different cells and tissues can have different thresholds between the biological and toxic effects of (poly)phenols, establishing the optimal in vivo doses that may prevent T2DM is an important issue. Further toxicity studies of (poly)phenols must be carried out to ascertain the optimal dosage for safe usage and to avoid the potentially adverse effects in humans.

An alternative strategy to overcome the metabolic transformation that may alter the antidiabetic effects of plant (poly)phenols is to develop new means to deliver these compounds, minimizing their degradation via the use of new engineered delivery systems (nanoparticles) [52]. A wide range of nano-formulations have been investigated and developed to try to maximize the effects of (poly)phenols. Nevertheless, further studies are still needed to demonstrate the efficacy of these systems to deliver the bioactive (poly)phenols to in vivo target tissues and clarify the concerns about the toxicological safety of these particles in the human body [53]. Animal models can also be useful for investigating the pharmacological administration of bioactive (poly)phenols through alternative routes (i.e., subcutaneous or intraperitoneal), as well as to understand the specific transformation, if any, and therapeutic effects of a given parent compound administered via these other routes. In a recent example, Chen et al. [54] investigated the regulatory effects on various neuropathy and inflammatory biomarkers following the injection (intrathecal and intraperitoneal) of epigallocatechin gallate (EGCG) using a rat model of diabetes. Nevertheless, this area of research is less developed, and there is still a long way to go to establish whether this strategy can be implemented for the medical delivery of (poly)phenols in humans.

## 5. General Discussion and Recommendations to Advance the Understanding of the Effects of (Poly)phenols on T2DM

It has been clearly established that the high heterogeneity of the studies and the variability of the results, as well as the difficulties related to the implementation of clinical intervention trials on the effects of (poly)phenols against T2DM, still pose a great challenge to attaining more definitive and demonstrative effects, which would eventually allow these compounds to be recommended for preventing the development of T2DM. Future intervention studies looking at the antidiabetic effects of the intake of (poly)phenols should focus on deciphering experimental variability and interindividual variability in response to these compounds, as well as the factors influencing this variability. This research should help us to understand the true responses to (poly)phenols and to progress toward the application of more effective nutritional recommendations [55]. Nevertheless, this is not a trivial task, and we appear to be a bit stuck at this point of the research; therefore, the implementation of new and better-quality RCTs should be encouraged. So, what can we do to try to improve this research in order to find sufficient evidence supporting the beneficial regulatory activity of (poly)phenols against T2DM?

An essential issue that still needs to be addressed is the identification of the individuals that would respond to the intake of these compounds. There appears to be some indication that patients with pre-diabetes or T2DM would benefit most from the intake of (poly)phenols. The evidence appears to be substantial in the particular case of some mixed compounds (e.g., *Moringa oleifera* extract) as well as some single compounds such as resveratrol. Therefore, more trials focused on the antidiabetic effects of these products in sufficiently large and well-characterized individuals with pre-diabetes and diabetes are encouraged. The sample populations of patients should also be as homogeneous as possible regarding other very important influencing factors, such as sex, age and body weight. Whether the effects of (poly)phenols improve those of anti-diabetic drugs remains to be demonstrated and should also be carefully investigated. The key issue is to use significantly larger sample populations that are narrowed down to more specific and well-characterized individuals (‘true responders’). It would be also of great interest, once the responders are identified, to further confirm their responses through additional trials and to extend the evidence to other individuals with the same phenotypic characteristics. An additional and critical issue that remains to be solved is the high diversity of the investigated test products (commonly mixed (poly)phenols) and different test materials (whole foods, encapsulated extracts, solubilized compounds), which are not usually completely characterized and are often tested against a control or placebo that is also insufficient to ascribe the potential results to the (poly)phenols. It is thus essential that, to definitively prove the antidiabetic properties of orally administered (poly)phenols, we improve the characterization of the test products by providing their full composition and testing them against fully described placebo products. These products should contain, as much as possible, the components of the mixture except for the (poly)phenols to be tested. An additional strategy might be to confirm the antidiabetic effects of a single (poly)phenol, and then to build up the evidence for that single compound as part of a synthetic mixture of compounds, and as a component of natural extracts and foods.

As it has been repeatedly stated throughout this review, it is also essential that we understand the association between the antidiabetic effects of a (poly)phenol or group of (poly)phenols and their metabolic fate. Identifying and quantifying the full set of host and microbial metabolites derived from the intake of (poly)phenols will help us to differentiate between individuals’ metabolism (metabotype), to identify the true potential bioactive molecules, and to understand the potential mechanisms of action underlying those effects. The current evidence of the association between the antidiabetic effects of orally ingested (poly)phenols and the circulating in vivo compounds and/or derived metabolites is very limited; therefore, more bioavailability studies are needed, especially those carried out in combination with bioactivity studies. In this review, we have indicated some good examples to follow in order to improve future animal and human studies.

Overall, it has been shown that the average effects of different dietary (poly)phenols on the main biomarkers of T2DM are small with reductions in a few mg/dL for FPG and a small % decrease for Hb1Ac. In addition, to statistically demonstrate the significance of these changes, it is important to understand the relevance of these small size effects in the context of the development of diabetes. It is essential that we confirm that the changes are maintained in the long-term consumption of these products with no adverse or toxic effects. Most of the limitations and problems indicated here can be partly overcome through a collective effort by the scientific community working in this area, with a common focus on filling research gaps that still exist. Through more homogeneous studies and data sharing among researchers, it is possible to accumulate the evidence supporting dietary (poly)phenols as a mean to combat T2DM. This was something widely discussed during the COST Action POSITIVe [56], but whether data sharing is truly taking place is not yet clear and should be thoroughly encouraged. Despite some leading research groups already working under improved pre-clinical experimental conditions, recently published in vitro and animal studies looking at the antidiabetic properties of (poly)phenols are unable to represent the in vivo conditions in humans. We need to continue spreading and reinforcing the need to apply new and improved models that will complement human trials and help to describe the true molecular and cell mechanisms of action triggered by the intake of (poly)phenols. In this review, we summarize some of the most recent evidence regarding the impact of the intake of (poly)phenols on T2DM. Some evidence of their regulatory benefits exists both for mixed extracts and single compounds. Nevertheless, there are still a number of issues that need to be improved to reach consistent and sufficient evidence to support the benefits of these compounds in specific populations. We highlighted some of the main issues to reinforce these findings, which we hope are increasingly implemented in future studies.

## Figures and Tables

**Figure 1 nutrients-14-03563-f001:**
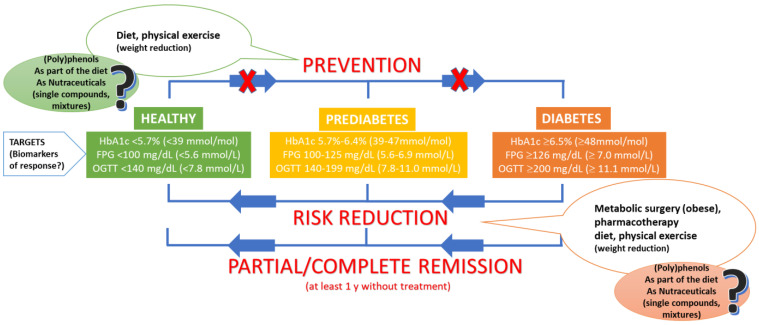
Summary graphics representing the current criteria for the screening and diagnosis of prediabetes and diabetes as well as the main means to combat this disorder. (Poly)phenols may be used both as dietary or nutraceutical compounds for the prevention and (or) treatment of the disease.

**Figure 2 nutrients-14-03563-f002:**
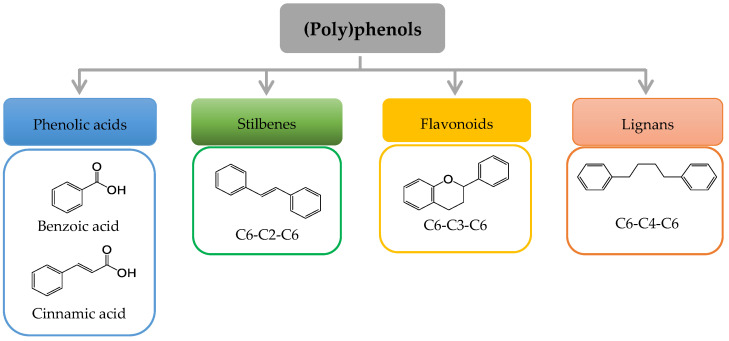
Basic structure of the major groups of (poly)phenols according to the number of phenol rings and the structural elements that bind theses rings.

## Data Availability

Not applicable.

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
