# Peer review of "Updated Information of the Effects of (Poly)phenols against Type-2 Diabetes Mellitus in Humans: Reinforcing the Recommendations for Future Research"

_nutrients, 2022, doi:10.3390/nu14173563_

Round 1

Reviewer 1 Report

I have two major concerns when it comes to this paper in terms of lack of merit of publishing the study:

1. Research on phenolic compounds and diabetes is vast and extensive. At this point, what would be helpful would be a systematic review based on MOOSE or PRISMA formats where all these papers can be analyzed methodically. What the authors have presented here is a literature review. As far as the research platform is concerned, it would be greatly beneficial as to set the direction based on gaps and voids, rather than regurgitation and summarizing existing literature.

2. The language used throughout the study is quite awkward. In fact, at some points, it is not clear as to the message which needs to be conveyed. There is a lot of polishing required in this aspect, for e.g. Introduction, line 1, 'Type 2 diabetes mellitus (T2D) remains a worldwide increasing disease' can be refined as 'Type 2 diabetes mellitus (T2DM) is on globally on the rise'. The assistance of a native English language speaker is required to rectify this issue.

Reviewer 2 Report

Menezes et al., have reported the role of polyphenols in the treatment of Type 2 diabetes. The authors have nicely written the review covering all the aspects of importance of polyphenols in the treatment of the disease. However it would be nice if the authors can add some benefits of Tea in the treatment of the disease as some plants are rich in polyphenols than others. 
